# *Cannabis sativa* Cannabinoids as Functional Ingredients in Snack Foods—Historical and Developmental Aspects

**DOI:** 10.3390/plants11233330

**Published:** 2022-12-01

**Authors:** Marlize Krüger, Tertia van Eeden, Daniso Beswa

**Affiliations:** 1Department of Life and Consumer Sciences, School of Agriculture and Life Sciences, University of South Africa, 28 Pioneer Ave, Florida Park, Roodepoort 1709, South Africa; 2Department of Biotechnology and Food Technology, Faculty of Science, University of Johannesburg, 25 Louisa St, Doornfontein, Johannesburg 2028, South Africa

**Keywords:** *Cannabis sativa*, cannabis edibles, cannabinoids, pediatric exposure, side-effects

## Abstract

The published health benefits of *Cannabis sativa* has caught the attention of health-conscious consumers and the food industry. Historically, seeds have long been utilized as a food source and currently there is an increasing number of edibles on the market that contain cannabis. Cannabinoids include the psychoactive constituent, delta-9-tetrahydrocannabinol (THC), and the non-psychoactive cannabidiol (CBD) that are both compounds of interest in *Cannabis sativa*. This paper looks at the distribution of nutrients and phytocannabinoids in low-THC *Cannabis sativa*, the historical uses of hemp, cannabis edibles, and the possible side-effects and concerns related to cannabis edibles. Several authors have pointed out that even though the use of cannabis edibles is considered safe, it is important to mention their possible side-effects and any concerns related to its consumption that negatively influence consumer acceptance of cannabis edibles. Such risks include unintentional overdose by adults and accidental ingestion by children and adolescents resulting in serious adverse effects. Therefore, cannabis edibles should be specifically packaged and labelled to differentiate them from known similar non-cannabis edibles so that, together with tamperproof packaging, these measures reduce the appeal of these products to children.

## 1. Introduction

Food products infused with chemical compounds that possess health benefits beyond basic nutrition [1] are piquing the interest of health-conscious consumers and the pharmaceutical and food industries. Such interest has intensified since cannabis was legalized for private use in some countries, including South Africa [2], and while the sale and distribution of high-content THC cannabis remains prohibited, some products that contain lower amounts of THC and other cannabinoids such as cannabidiol (CBD) have either been lowered or removed as a scheduled drug [3]. Among such products are cannabinoid-infused food products, beverages and oils, also known as cannabis edibles, that can exert their beneficial effects by modulating human cannabinoid receptors [4]. 

Cannabis is a generic term describing an annual herbaceous plant that belongs to the *Cannabaceae* family [5,6] with *Cannabis sativa*, *Cannabis indica* [7] and, though still debated, *Cannabis ruderalis* recognized as the three main species in cannabis [8]. *Cannabis sativa* contains an abundance of non-nutritive, bioactive phytochemicals known as phytocannabinoids, with delta-9-tetrahydrocannabinol (THC) and cannabidiol (CBD) as phytocannabinoids of particular interest. *Cannabis sativa* plants containing less than a specified amount of THC in dried weight are also known as hemp. In certain European countries, THC levels should not exceed 0.2%. In the United States of America (USA) [9,10,11] and Canada [12], THC levels of up to 0.3% are acceptable while the allowable limit is up to 1% in Switzerland [13]. *Cannabis sativa* plants containing more than the specified THC limit (high THC or marijuana) are considered as schedule 1 drugs [14,15]. Therefore, high-THC cannabis is categorized with substances that are defined as having the “highest potential for abuse”, “no currently accepted medical use”, and as not having “accepted safety for use under medical supervision” [16].

Those who use the aerial parts of the marijuana plants, such as the leaves and flowers, became intoxicated after smoking and when smoking on a regular basis, may have a notable increase in the incidence of health risks such as mental illnesses [17]. As a result, these plants were perceived as criminal and unacceptable to communities because most consumers could not differentiate between psychoactive and non-psychoactive cannabis plants. In contrast, the use of cannabis plants with low levels of THC in medicine and foods is based on various, potentially beneficial cannabinoid compounds [18] including CBD [5]. The use of CBD may effectively treat a wide range of diseases and disorders including epilepsy, neurodegenerative diseases, neuropsychiatric disorders and rheumatic diseases [19]. Plants are also used in other industries such as textiles, building material [20], ornamental [21], biomedical and food industries [22]. This paper reviews low-THC *Cannabis sativa* (or hemp plants) and associated cannabinoid CBD as a functional ingredient in food products.

## 2. Distribution of Nutrients and Phytocannabinoids in Low-THC *Cannabis sativa*

### 2.1. Nutrients

Many consumers are uninformed regarding the nutritional benefits associated with *Cannabis sativa*, or hemp, plants, despite a considerable amount of scientific proof [9,23,24]. Figure 1 indicates that the stems, seeds, roots and flowers of the hemp plant are a substantial source of nutrients (fat, fiber, ash, protein and vitamins). In addition, Figure 1 indicates the relative amounts of phytocannabinoids such as delta-9-tetrahydrocannabinol (THC), cannabidiol (CBD) and cannabigerol (CBG) that could be found in the respective parts of the plant [23,25,26]. The most common parts of the hemp plant eaten by humans are the seeds that contain 25 to 35% total lipids, and proteins ranging between 20 and 30% that are highly digestible with substantial amounts of essential amino acids, as well as 20 to 30% total carbohydrates, an ash content of 3.7 to 5.9% and 25 to 28% total dietary fiber [26,27,28,29].

Proteins found in the hemp seed are legumin-type globulin edestin (67 to 75%) and globular-type albumin (25 to 37%) [9] but, like most plant proteins [30], the seeds are low in the essential amino acid, lysine [9]. Many of the health-promoting properties of these seeds are ascribed to their high levels of essential fatty acids (EFA) such as linolenic and linoleic fatty acids, and polyunsaturated fatty acids [27,31,32].

### 2.2. Phytocannabinoids

Around 110 different cannabinoids have been identified in cannabis [18,33]. Cannabinoids are mostly concentrated in the glandular trichomes (hairy outgrowths) of flowering plants, liverworts, and fungi [34]. Thus, hemp plants contain many non-psychoactive cannabinoids such as cannabidiol (CBD), cannabigerol (CBG), and cannabichromene (CBC) along with other non-cannabinoid compounds belonging to various classes of natural products [35]. CBD is abundantly distributed across the different parts of the hemp plant, with leaves having the highest (2%) followed by the stem (1.8%) and the seeds having the lowest concentration (0.02%) [26]. Some of these cannabinoids may be used to treat human ailments such as pain, anxiety and cachexia [34,36], to stimulate appetite, and to act as an antiemetic [37]. The primary metabolites (amino acids, fatty acids, and steroids) in low-THC hemp plants are used to synthesize secondary metabolites such as flavonoids, terpenoids, lignans, alkaloids and phytocannabinoids [38], the latter being characterized by a C21 terpenophenolic backbone [18,33]. The synthesis of THC involves the decarboxylation of delta-9-tetrahydrocannabinolic acid (THCA); whereas CBD is a decarboxylated product of cannabidiolic acid (CBDA) [39].

## 3. Uses of Hemp

Archaeological findings indicate that cannabis has been cultivated for over 6000 years, possibly originating in China [40]. The most popular use of cannabis has involved the psychoactive variant (*Cannabis sativa* subsp. *indica* or marijuana) where humans smoke a dry combination of its shredded leaves, stems and flower buds [41]. This has often led to behavior and health disorders (cannabis use disorder) [42] consequently leading consumers to have negative associations with cannabis as a whole [43]. Figure 2 indicates the various parts of hemp plants that have been used to produce a wide range of products including hemp fiber for clothing, rope, paper and medicines [5,11,20,44]. In addition, cosmetics and detergents were prepared from crude extracts of hemp leaves [6,20]. Hemp seeds were used to provide lamp and cooking oil and as a food source [25,44,45,46], which was considered to be of high nutritional value [25,47].

The hemp seed is achene [24]: containing only one seed that nearly fills the pericarp and does not split open when the seeds are ripe [48]. The hulled seed is greenish grey in color [49] with a pleasant, nutty taste [50]. It is oval or spherical in shape with green spots due to plant tissue beneath the epidermis [24]. Figure 3 shows the unshelled seed as well as the shelled seed with green specks. 

The edible forms of hemp seeds include hulled seed, hempseed oil and its crushed by-products, hemp seed cake, or meal [53]. Hemp seeds, including the hull, were eaten in ancient times for their nutritional and medicinal benefits [47]; whereas, the removal of the hull is considered a recent development [50]. Hemp plants provided for humans who harvested their seeds as a grain either in a meal form (ground), roasted whole, or cooked into porridge [54]. These seeds were also consumed to treat joint pain, malaria, and memory loss [55,56].

More recently, hemp seeds have proven to be a versatile ingredient in food production [57] with the dehulled hemp seed and/or its by-products being successfully incorporated into bread, dairy, meat and energy bars to enhance their nutritional value and sensory properties [9]. When incorporated into food formulations, hemp seeds can impart beneficial bioactive substances that reduce the risks of cardiovascular disease, diabetes and cancer, and improve gut and immune function [58]. In addition, the inclusion of hemp oil or meal in animal feed has shown promise by increasing animal milk yield and nutritive quality [59], and its incorporation increases the stability of egg yolks [43].

Cannabis arrived in Africa about 1000 years ago and has become an important source of income for rural communities [60]. However, there are limited publications that show how cannabis has been used as a food in Africa. Legalization of the use of cannabis has shifted both low-THC and high-THC cannabis practices from the black market to the legal market where more opportunities for innovation exist. Moreover, acceptance of the value of cannabis is increasing in both scientific and public circles [61], with CBD edibles as the preferred method of consumption [62]. This has resulted in the commercial and domestic development of cannabis edibles such as baked goods, sweets, lozenges, and beverages [63]. This range reflects a growing interest from some major manufacturers in exploring the benefits of hemp-based ingredients to enrich their existing and new food products. This strategy is feasible as most of the current cannabis edibles manufacturers only infuse health-benefitting cannabis extracts into their existing product formulations [25] (Figure 4).

## 4. Cannabis Edibles

The concept of nutrition has reached a greater audience because of reports describing diet- and lifestyle-related non-communicable diseases [24]. As a result, health-conscious consumers are moving towards a plant-based diet and this shift influences the demand for plant-based proteins such as those found in hemp seed [63]. Several studies reported that the inclusion of hemp seed meal in dough increased the protein and iron content, essential amino acids (lysine, phenylalanine, histidine), fiber, iron (Fe) and zinc (Zn) [27,64] to contribute towards the palatability and color of the resulting gluten-free bread [65] and crackers [66]. 

Apart from the seeds, the inclusion of cannabinoids such as CBD in functional foods could support health and reduce the risk of disease. Food products that contain cannabinoids such as CBD are referred to as cannabis edibles and include a range of products from baked goods and sweets to beverages [25,67]. Though not preferred to smoking, ingesting cannabis has always been a popular method of cannabis consumption [68,69,70,71,72]. Some of these products are indicated in Figure 5. Various snack foods that have been infused with CBD are available in the United States and they include jelly sweets [73], carbonated drinks [74] and snack bars [75]. The names of these products such as “Chill”, “Relax” and “Recess” often hint at the therapeutic effects associated with CBD and the number of countries that permit over-the-counter or online access to CBD products is on the rise [76]. However, only one product, intended for medicinal use, containing CBD has been approved by the FDA for human consumption (Epidiolex^®^) and claiming CBD as a dietary supplement is currently illegal in the United States [77].

Edibles usage is classified into medicinal and recreational purposes [63]. About 16–26% of health-conscious patients consume cannabis edibles for medicinal purposes [78] but there is a scarcity of data available on the usage of cannabis for recreational purposes. Globally, biscuits, cookies and candies are popular ready-to-eat snacks consumed across age groups. Consequently, the manufacturers of cannabis edibles available in South Africa have taken advantage of these snacks by infusing their formulations with cannabis extracts [79].

**Figure 5 plants-11-03330-f005:**
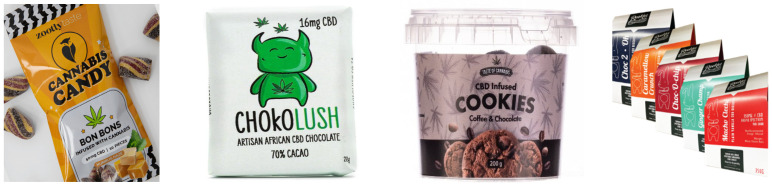
Some cannabis edibles on the South African market [79].

South Africa permits some general, low-risk health maintenance or health enhancement claims on sold CBD products. However, the permitted amount of CBD per serving should not exceed 20 mg and the sales pack may not exceed a total of 600 mg of CBD [3] and should contain less than 0.001% THC [80]; whereas, in the USA, edibles should contain less than 0.3% THC and CBD at levels between 10 and 1500 mg [81]. Despite these regulations, there are concerns about some errors in labelling where the cannabinoid concentration of the edibles does not correspond with the content recorded on the labels [38]. Some edibles have been found to contain either less or more cannabinoid content than the amount claimed on the label [82]. Figure 5 indicates some of the edibles available on the South African market, containing between 16–150 mg CBD per sales pack and 5–20 mg CBD per serving [79]. Yet, many consumers remain skeptical about trying cannabis edibles and the next section will elaborate on the possible side-effects and concerns related to the ingestion of hemp.

Just as smoking cannabis may have side-effects, consuming edibles may also have pharmacokinetic effects [63] whose symptoms may be delayed as ingested cannabis needs to be metabolized before being absorbed in the intestines. Such ingestion depends on factors such as previous meals [83], metabolism and body weight, as well as the concentration and type of cannabis or cannabis product [84,85], and can take up to several hours to affect the brain [83,86]. Edibles, therefore, have an increased likelihood of causing an unintended high as the user is often impatient and may consume more of the edible before feeling the effects [86,87,88]. Thus, CBD, taken in doses in excess of 1500 mg, may exert side-effects [89].

## 5. Possible Side-Effects and Concerns Related to Cannabis Edibles

Cannabidiol (CBD) and CBD-products for private use are increasing in popularity amongst European [90], South African [91], American and Canadian consumers [92]. Accordingly, it is necessary to list some medical and social concerns relating to the ingestion of cannabis edibles. Due to the novelty of CBD, there has not yet been a validated method to assess and verify purity and CBD content [93] and claims on CBD products are currently unregulated and unverified [94]. This uncertainty regarding risks has resulted in only one CBD product being approved by the US Food and Drug Administration (FDA) [95] particularly because CBD is not yet classified as a food supplement nor as a medical drug [96]. 

Secondly, there is concern regarding either intentional or accidental adulteration or contamination of cannabis edibles that might be harmful to the consumer, including heavy metal and pesticide contamination [97], contamination by toxic residual solvents used to extract CBD [98] or THC, and synthetic cannabinoid contamination of CBD [93,96,99,100]. In addition, relatively little is known about the chronic and hormonal effects following ingestion of CBD or its interactions with other drugs [93,95,101]. In general, there are considerable knowledge gaps amongst consumers as well as medical professionals regarding CBD dosage [102,103], possible side-effects or when to seek medical assistance [93,104]. Interestingly, the FDA has only approved CBD with THC levels of less than 0.3% for use in cosmetic products [105,106] and intoxication from consuming cannabis edibles may vary from person to person [105] who present with various symptoms [63].

A major concern surrounds pediatric exposure to cannabis edibles [107]. A study [108] reported that around 46% of Canadians were willing to try cannabis edibles but older respondents expressed concern about the health risks for children in households in which cannabis edibles were consumed [109,110]. These concerns are strengthened by the fact that most cannabis edibles are shaped and packaged in a manner that mimics known edibles such as cookies, brownies, gummies and candy. Thus, children may not know the difference between cannabis-infused and normal edibles and naturally assume that cannabis edibles are ordinary treats [107]. As a result, various cases have been reported regarding unintentional exposure of cannabis edibles to children [41,42,108,110,111].

With this range of concerns regarding the ingestion of cannabis edibles from production lines that may have lax quality assurance measures leading to the production and sale of cannabis edibles with unwanted constituents, it is essential that legislation prescribing cannabis product preparation and labelling should be clear and enforceable [93].

## 6. Conclusions

*Cannabis sativa* possesses many health-promoting qualities and so it has played an effective role as a traditional medicine to treat a variety of ailments from pain, anxiety and weight gain through to conditions such as cardiovascular disease and diabetes, as well as infectious diseases such as malaria, and cancer. Opinion regarding cannabis edibles is changing amongst consumers and most countries around the world are shifting towards the legalization of the recreational and medicinal use of cannabis leading to a rapid increase in the global acceptance and availability of cannabis edibles. A wide range of products infused with cannabis extracts are currently available on the market and, unfortunately, this increases the possibility of side-effects to the consumer. Not only are most cannabis edibles not regulated, but there is risk of accidental ingestion of cannabis, particularly by children. Such incidences are on the rise in those countries where cannabis has been legalized or decriminalized. Most cannabis edibles have not been approved by the FDA but are currently subject to evaluation regarding long-term safety and cumulative effects in humans. This calls for efficient and effective safety measures to prevent pediatric cannabis poisoning and accidental unintended high when amending policies for cannabis usage. In addition, there must be some form of commercial and consumer awareness regarding the optimal preparation and packaging of cannabis edibles to promote their safe and enjoyable consumption.

## Figures and Tables

**Figure 1 plants-11-03330-f001:**
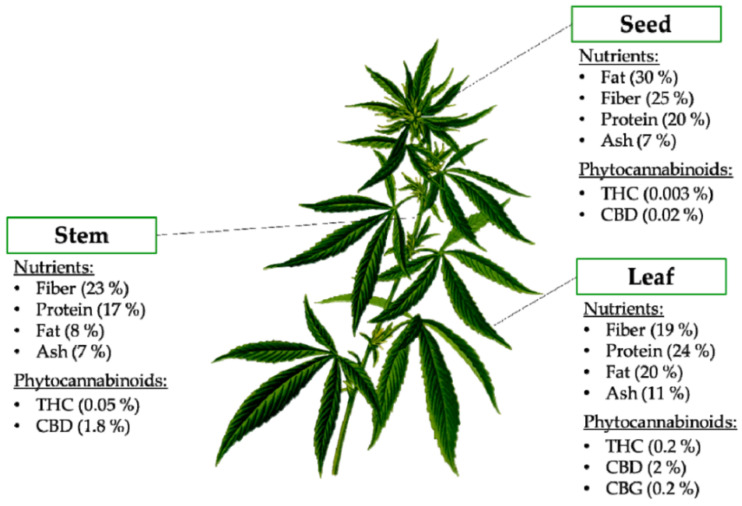
Nutritional composition and phytocannabinoids present in different parts of the hemp plant [26]. Abbreviations: CBD, cannabidiol; CBG, cannabigerol; THC, delta-9-tetrahydrocannabinol.

**Figure 2 plants-11-03330-f002:**
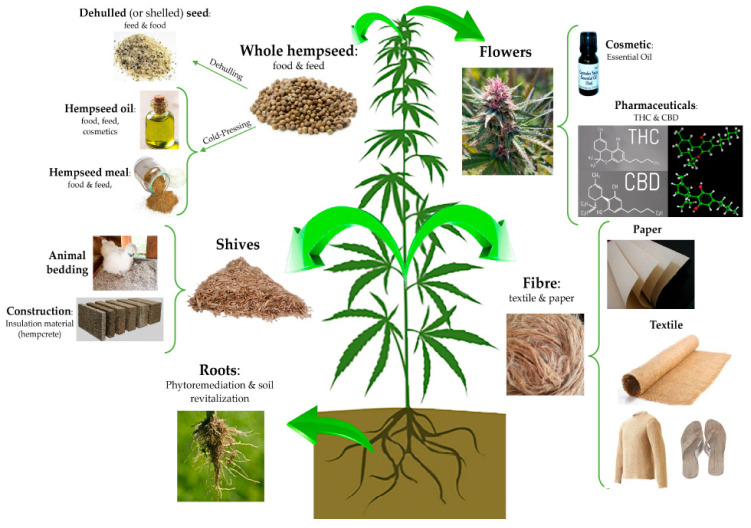
Various uses of the hemp plant [44].

**Figure 3 plants-11-03330-f003:**
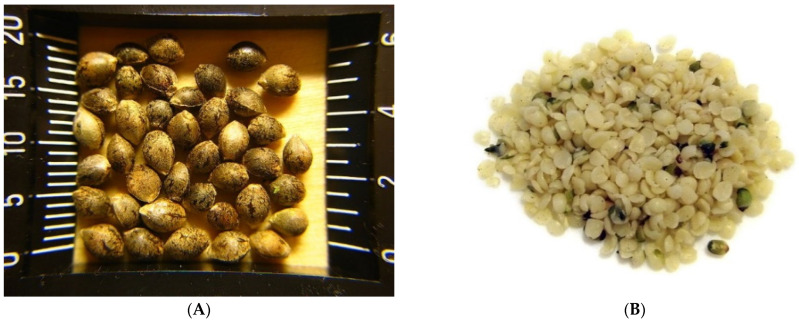
(**A**) Unshelled hemp seeds [51]; (**B**) Hulled hemp hearts [52].

**Figure 4 plants-11-03330-f004:**
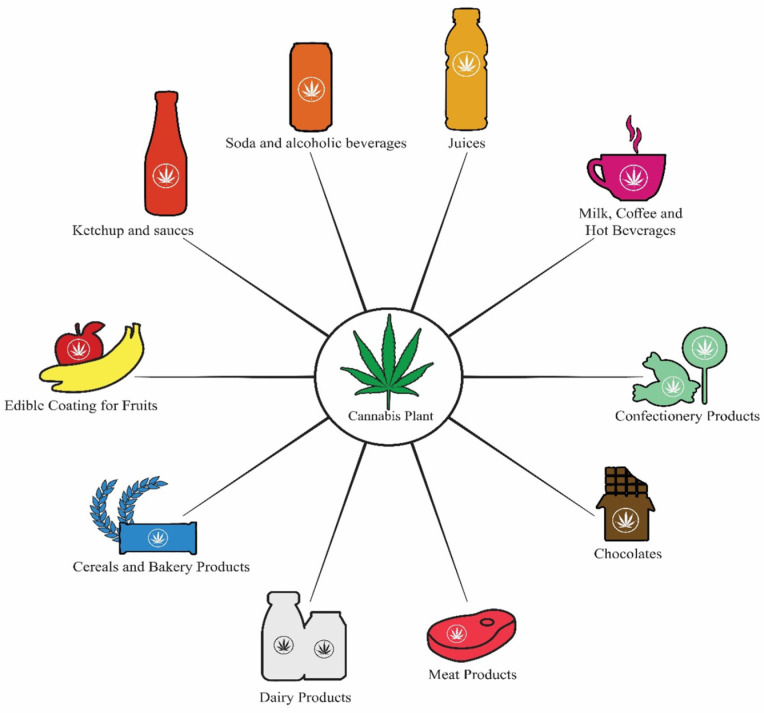
Various cannabis edibles in the food and beverage industry [25].

## Data Availability

Not applicable.

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
