# Peer review of "Cannabis sativa Cannabinoids as Functional Ingredients in Snack Foods—Historical and Developmental Aspects"

_plants, 2022, doi:10.3390/plants11233330_

Round 1

Reviewer 1 Report

Summary:

This a narrative review of cannabis as a food source, including historical aspects of cannabis use and consumption and current issues in cannabis edible regulations. This is an interesting topic, though the manuscript would benefit from some changes to the structure and more consistency in language.

Major Points:

1.     I honestly found this manuscript hard to follow. This is partly because it’s not clear what are the exact purpose and scope of the manuscript. Is the focus on non-psychoactive cannabis foodstuffs? It seems like maybe yes, since there’s a lot of focus on CBD and preparations of cannabis with low/no THC. In that case, maybe the authors should mention THC and other possibly psychoactive cannabinoids in the Introduction, but make it much more clear that this manuscript is not meant to review the use of psychoactive cannabis edibles. Then, reorganize the manuscript a bit – as an example, the authors could provide an overview of the cannabis plant (i.e., move section 4 to be immediately after the introduction), then discuss historical perspective of cannabis use for industrial and nutritional uses (combine sections 2, 3, and 5), and then from there focus only on non-psychoactive cannabis edibles (sections 6 onward).

Minor Points:

2.     While I understand the authors are using the term “marijuana” to refer to the psychoactive preparations of cannabis and distinguish from less psychoactive “hemp” preparations, the word “marijuana” has a history of being used to propagate racist stereotypes and the cannabis field in general is moving away from using this term. In my opinion, the authors should not use the term “marijuana” and instead refer to low-THC and high-THC cannabis cultivars.

3.     The last sentence on the first page starting “Those who use leaves of marijuana as tobacco…” is confusing to me. What does it mean to “use leaves of marijuana as tobacco”? Do the authors just mean “smoke”?

4.     Lines 54-55 – what are “active, healthy cannabinoids”? By active, do the authors mean psychoactive or pharmacologically active, or something else? What makes a cannabinoid “healthy”?

5.     Lines 55-57 – I would not say THC is “known” to alleviate neuropathic pain, etc. THC certainly has analgesic potential, but its efficacy in human trials is mixed, so I feel this statement is misleading.

6.     Lines 70-74 feel out of place to me. I think the authors should have a separate paragraph to discuss historical use of cannabis for recreational purposes and discuss potential harms of high-THC cultivars of cannabis separately from the paragraph about hemp.

7.     Figure 3 – there are percentages associated with phytocannabinoids in the different parts of the cannabis plant. I think it would be helpful if the authors could clarify what these percentages actually represent (e.g., are they just meant to demonstrate relative abundance of cannabinoids in different parts of the plant?). Phytocannabinoid composition differs quite dramatically between different cultivars of cannabis.

Author Response

Thank you for your valuable review, your comments and suggestions significantly helped us in improving the content of our manuscript.  

Reviewer 2 Report

In the current manuscript, Krüger et al. have reviewed Cannabis-derived cannabinoids as food functional ingredients. Although the topic is attractive, there are some concerns that should be addressed.

-There are some typographical and grammatical errors.

-The paper title is well stated; it is informative and concise.

-Abstract is well structured.

-The introduction was not well written, and it is too briefly presenting the subject and research problem. For instance, lines 39-41, Cannabis is a multipurpose plant, therefore, "...they were used for medicinal purposes." should be changed to "...they were used for various purposes such as industrial (10.3906/bot-1907-15), ornamental (https://doi.org/10.3390/plants11182383), and pharmaceutical (https://doi.org/10.1007/978-981-16-8822-5_4) applications."

-Section "8. Consumer acceptance of cannabis edibles" should be improved.

-In general, conclusions are correct, but not sufficient. The conclusion should be improved.

Author Response

(The authors gave the same response as above.)

Reviewer 3 Report

I cannot recommend acceptance of the paper in the present state since this article does not add anything new to the knowledge base.

Specific comments and suggestions are given below.

Line 13: Please rephrase ''increasing number of cannabis edibles, known as cannabis edibles….''

Lines 39-42: Please rephrase the explanation of classification: it must be understandable for the readers that hemp and marijuana belong to the same Cannabis sativa plant species, differing botanically at a subspecies level as C. sativa subsp. sativa (hemp) and C. sativa subsp. indica (marijuana).

I propose to make a clear distinction between hemp and marijuana throughout the manuscript.

For hemp, I also propose to use C. sativa subsp. sativa for hemp or just hemp throughout the manuscript.

It is quite unclear when you are referring to hemp and when to THC-reach plant.

Also, it will be good to mention that cannabis is a general word that refers to all plants that belong to the Cannabis genus.

Line 44: For smoking. the fruiting and flowering tops and leaves are used, not only leaves

Line 59-60: CBD cannabinoid, not CBD cannabinoids, I suppose.

Line 73: Please add the reference.

Line 77: Cannabis seeds or hemp seeds?

Line 104: Please add the full name for CBG.

Lines 104-105: ‘’ The only part of the plant eaten by 104 humans is the seeds’’. What about tea? Isn’t it the mixture of seeds, flowers and leaves?

Line 154: Data on cannabinoid content in edibles is missing. Also, statistics on use of edibles would be useful.

Author Response

(The authors gave the same response as above.)

Round 2

Reviewer 1 Report

The authors have done a great job responding to reviewer comments and all of my concerns have been addressed. I appreciate the effort the authors took to incorporate the feedback.

I do have one new comment - the authors refer to cannabis edible "overdose" on 97, line 219. I would avoid the use of the term overdose in reference to cannabis, and instead use a term like "unintended high". This is just a recommendation though, the authors can keep the phrasing as is if they feel strongly about the word overdose here. There's some debate in the field whether cannabis "overdose" is an appropriate term, but there's no universally accepted answer.

Author Response

Thank you for adding value to our manuscript.

Reviewer 2 Report

Unfortunately, my comments have been not addressed. The authors should address the following comm

Lines 50-65: wrong introduction.  There are lots of drug-type genotypes derived from C. sativa subsp. sativa. This paragraph should be completely revised. The differences between hemp and drug type are based on the THC level, and it varies across different countries (for example cannabis with more than 0.3% THC is considered a drug type in North America)

As previously mentioned Cannabis is a multipurpose plant. The authors should mention in the introduction that cannabis has been used for various purposes such as industrial (10.3906/bot-1907-15), ornamental (https://doi.org/10.3390/plants11182383), and pharmaceutical (https://doi.org/10.1007/978-981-16-8822-5_4) applications.

Lines 175-206: Cannabis edibles: it is superficial. this section should be improved.

Author Response

Thank you for reviewing our manuscript, we highly appreciate your valuable input and suggestions. 

Reviewer 3 Report

Data on cannabinoid content in edibles is still missing. Section 4. Cannabis edibles must be improved.

Author Response

Thank you for your valuable input and suggestions.  

Round 3

Reviewer 2 Report

My comments have been addressed. I think that the current form of the manuscript is suitable for publishing in the Plants.